# Non-Destructive Determination of Alkylresorcinol (ARs) Content on Wheat Seed Surfaces and Prediction of ARs Content in Whole-Grain Flour

**DOI:** 10.3390/molecules24071329

**Published:** 2019-04-04

**Authors:** Jiahuan Wang, Xin Gao, Zhonghua Wang

**Affiliations:** State Key Laboratory of Crop Stress Biology for Arid Areas, College of Agronomy, Northwest A & F University, Yangling 712100, China; wangjiahuan327@126.com

**Keywords:** alkylresorcinols, wheat seed, lipids, GC-MS, whole grain, correlations analysis

## Abstract

Alkylresorcinols (ARs) are beneficial for human health and can be used as biomarkers for whole-grain cereal intake. In previous studies, ARs content in whole-grain flour was determined by milling the seeds into powder, thus destroying their structure. In this paper, we adopted a non-destructive extraction approach. Chloroform and ethyl acetate extraction methods were carried out to extract lipids from the seed surface and whole-grain flour of 36 wheat varieties in China, respectively. GC-MS analysis identified chemical compounds in which ARs were the major compounds for all the samples. The average total content of ARs (624.223 µg/g) in whole grains was much higher than that on seed surfaces (4.934 µg/g), with a positive correlation (r = 0.863, *p* < 0.001) between these two parameters. The results suggested that the total ARs content on the seed surface can be used to predict their content in whole-grain flour. Without affecting the seed germination rate and damaging seed structure, we demonstrate that a non-destructive extraction approach is an appropriate and useful method, facilitating the development of rapid screening methods suitable for use in wheat breeding.

## 1. Introduction

Alkylresorcinols (ARs), also called 5-*n*-alkylresorcinols, are present in large amounts in cereals [1,2,3]. Among the common crops used for human consumption, wheat (*Triticum aestivum* L.) generally has a high ARs content with odd-numbered homologues from C15 to C25 [4,5]. ARs are of interest in nutrition science, which have been suggested as markers for whole-grain wheat in food and as biomarkers for human intake of whole-grain wheat [6,7,8,9].

The positive health effects of whole grains have been studied extensively in recent years, and whole grain intake has been consistently linked to decreased risk of developing type 2 diabetes [10,11,12]. Emerging in vitro data suggest that ARs may have bioactivities related to enzyme inhibition [13]. Previous reports also suggested that ARs could inhibit low-density lipoprotein oxidation, prevent cardiovascular diseases, and increase fecal cholesterol excretion [14,15,16]. Scientific organizations believe that humans would benefit from increased consumption of whole grain foods, as they can reduce the risk of developing chronic diseases [17,18,19,20]. Being able to breed wheat that is rich in ARs, with enhanced health benefits, is of great importance.

A substantial amount of direct and indirect evidence indicates that the majority of ARs are located in an intermediate layer of the caryopsis (including the testa, hyaline layer and inner pericarp), while fine wheat flour contains essentially little ARs [1,21,22,23]. Therefore, to determine cereals’ ARs content, they are always milled into powder [23,24]. To date, there have been many methods reported for extracting ARs from whole grain flour and bran. In most cases, these extraction methods tend to require long extraction times (6 to 24 h) [25,26,27], complicated operations, and high chemical-analytics costs [28]. After extraction, cereal samples were not recycled, and thus the materials were wasted. It is important to note that, screening and breeding of wheat varieties with high AR content tends to be one of the major wheat breeding targets [19,29]. In breeding studies, it is necessary to screen large numbers of lines; however, during the early stages of the breeding process, only small quantities of grain are available [29]. However, the lack of a non-destructive, simple-operation, cost-effective, and time-saving extraction method for the AR content determination limits trait selection in wheat breeding programs. Therefore, developing a non-destructive method that can help breeders efficiently and quickly analyze ARs in large populations of wheat materials is necessary.

The surface of above-ground plant organs is covered with a cuticle, a protective lipid structure sealing the tissue against the environment. The surface lipid serves as the major barrier preventing nonstomatal water loss, protects plant surfaces against pathogens and ultraviolet radiation, and affects plant-insect interactions [30,31,32]. However, seed surface lipid (SSL) extraction methods are rarely reported in wheat, and there has been little focus on the chemical composition of lipid on wheat seed surface. Lipids extraction from tissue surface using chloroform has been extensively used in many studies [2,33,34,35]. Furthermore, the chloroform extraction method is fast, simple in operation, and highly-efficient [36,37]. Previous use of chloroform (with 30-s dropping in chloroform) to extract surface lipids (including ARs) from wheat and rye leaves also highlights the method’s possible use for intact ARs extraction from other wheat organs [33,37]. To the best of our knowledge, there have been no reports on the relationship between the SSL and whole grain lipid (WGL). In particular, little is known about the correlations of ARs and AR homologues on seed surface and that in whole grains.

The aim of this study was to investigate the chemical composition of the SSL and WGL from 36 cultivars of wheat in China, and develop a non-destructive method for AR content analysis on seed surface and whole-grain flour. Therefore, both chloroform and ethyl acetate were tested for extracting lipids from wheat samples. At the same time, the relationships between the content of ARs and AR homologues on seed surfaces and in the whole grains were studied in detail. In addition, wheat seed germination tests were carried out to confirm that it is a non-destructive method. In this study, we provide a non-destructive method for efficiently predicting total ARs content in whole grains, to help breeders select wheat varieties (lines) with high ARs content quickly for future breeding studies. 

## 2. Results

### 2.1. Identification and Quantification of SSL Components

In order to clarify the optimal extraction time for the chloroform extraction method, different extraction times (10 s, 30 s, 60 s, 90 s, 120 s, 180 s, 240 s, 300 s, 600 s, 1800 s) were tested. A 4-min (240 s) immersion in chloroform was sufficient to yield more the 99.4% (*w*/*w*) of total surface lipids from wheat seed surface (Figure 1). Individual lipid components were also found fully extracted after 4 min (Figure 1). Therefore, this procedure was used for all wheat seed analyses, and the lipid material recovered by this procedure was collectively termed SSL. We then used this chloroform extraction method to examine the presence of lipids on seed surfaces from 36 wheat cultivars (Appendix B
Table A1).

GC chromatograms revealed that the SSL comprise of 25 chemical compounds (Figure 2a). The mass spectra of the ARs and *n*-methylalkylresorcinols (MARs) showed Mclafferty rearrangement fragments at *m*/*z* 268 (Figure 3a) and *m*/*z* 282 (Figure 3b), respectively. The peaks with these characteristic *m*/*z* values were considered resorcinolic derivatives [1,33,38]. 

The average total content of SSL in wheat varieties (n = 36) grown in China was approximately 16.16 μg/g. The mixtures contained large amounts of ARs (4.93μg/g) together with MARs (2.53 μg/g), alkanes (4.35 μg/g), and primary alcohols (PAs) (1.49 μg/g), 

The composition of wheat seed surface lipids was very distinct from that of wheat’s other parts [29]. AR homologous (C19–C25) were confirmed in SSL; they were identified by their molecular ion peaks at *m*/*z* 464 (AR 15:0), 492 (AR 17:0), 520 (AR 19:0), 548 (AR 21:0), 576 (AR 23:0) and 604 (AR 25:0) (Appendix A and Figure 4a). Similar characteristics have been reported in the literature for MS fragments [38]. ARs represented the major class of compounds in SSL (4.93 μg/g, ~34% [*w*/*w*]), which was somewhat lower than previously reported values for the outer pericarp of the wheat seeds (15 μg/g), which could be attributed to the latter using different extraction methods and wheat materials [21]. The data showed that AR 23:0 (1.93 μg/g) was dominant and AR 19:0 was the least abundant (Figure 4a). 

Interestingly, a series of MAR homologous (C19–C25) were identified, and confirmed by their molecular ion (Appendix A), which appeared at *m*/*z* 534 (MAR 19:0), 562 (MAR 21:0), 590 (MAR 23:0), and 618 (MAR 25:0). Similar characteristic MS fragments have been previously reported in *Chenopodium quinoa* [38] (Figure 5b), with MAR 21:0 being the most prominent (Figure 4a). Despite this abundance, no detailed analysis of the MAR components has been performed on wheat seed surfaces, and MARs have only been reported previously for wheat leaves [33].

A series of alkanes (C25–C33) were also identified in the SSL, with a relatively broad chain length distribution, and a strong odd preference for chains with an odd number of carbon atoms. C29, C27 and C31 were the major homologues of alkanes, and other alkanes were identified in lower concents. PAs and fatty acids were present as homologous series of predominant compounds, with an even number of carbons, both found ranging in chain lengths from C22 to C28 (Figure 4a).

### 2.2. Identification and Quantification of WGL Components

In the present study, we used ethyl acetate extracts lipids from whole grain samples for GC-MS analysis. The GC chromatograms showed that WGL contained 23 chemical compounds (Figure 2b). The average total WGL content in wheat varieties (n = 36) grown in China was approximately 1087.7 μg/g. The mixture contained a large amount of ARs (624.22 μg/g, ~57% [*w*/*w*]) together with sterols, monoacylglycerols (MAGs), stanols, alkanes, PAs, and MARs (Figure 4b). AR homologous (C15– C25) were identified in WGL, which was in accordance with previous reports [4,5]. The most abundant AR was AR 21:0 (300.8 μg/g), and AR 15:0 (8.0 μg/g) was the least abundant (Figure 4b).

A series of MAGs (~10.6% [*w*/*w*]), including α/β MAG 16:0 and α/β MAG 18:0 were identified (Figure 5c,d), with α-MAG 18:0 being the most abundant (Figure 4b). For α-MAG isomers, cleavage of the C(b)-C(g) glycerol backbone bond usually produces the base peak [M-CH2OTMSi]^+^ [39]. *m*/*z* 371 corresponds to this cleavage of α-MAG 16:0 (Figure 3c), while this ion is very weak in β-MAG 16:0 (Figure 3d). The most diagnostic ions of β-MAG isomers are [M-RCOOH]^+^ and [M-RCOOH-OTMSi]^+^ [39],which correspond to *m*/*z* 218 and 129 (Figure 3d), respectively. The total content of MAGs in WGL accounted for 159.3 μg/g of the materials, the α-MAG isomers being more prevalent than β-MAG isomers. Sterols (including cholesterol, campesterol and sitosterol) (Figure 5e–g) were also found in WGL at relevant contents, accounting for 214.94 μg/g, with sitosterol (156.2 μg/g) being the major compound (Figure 4b). Stanols (including campestanol and sitostanol) (Figure 5h,i) could also be identified (Figure 4b).

Furthermore, a series of alkanes, PAs, and MARs were also identified, although they were present in lower amounts (16.7 μg/g, 19.0 μg/g, and 6.1 μg/g, respectively). The numbers of homologues in these series was also much lower than that in SSL (Figure 4b). All the above results showed that ARs, sterols, and MAGs were the major compounds in WGL, especially ARs, which were the most abundant.

### 2.3. Differences in ARs Concents and Homologue Compositions on Seed Surfaces and in Whole-Grain Plour from 36 Wheat Cultivars

In our study, ARs were found in all 36 wheat samples. Total ARs and AR homologues content on seed surface and in whole grains from 36 samples grown in China during the 2017 crop year, are shown in Table A2 and Table A3. Total AR content in whole grains for all investigated wheat samples varied greatly from 407.0 µg/g to 872.3 µg/g (mean = 624.22 µg/g) (Table A3). These values were >120 times those of seed surface (ranging between 0.84 and 8.87 µg/g, mean = 4.93 µg/g) (Table A2), and these were consistent with other reports on wheat samples (350–900 µg/g) [1,4,6,21,24,40,41]. As shown in Table A2, total AR content differed among different wheat cultivars, but AR 21:0 and AR 23:0 amounts were consistently the highest, followed by AR 25:0 and AR 19:0, which were different from those of whole grain samples (Table A3). In whole grain flour, AR homologues content had the following order 21:0 > 19:0 > 23:0 > 25:0 > 17:0 > 15:0 for all samples (Table A3). These findings are in agreement with results reported by Ross et al. [42]. In addition, the content of the total ARs and their homologous on seed surface varied dramatically among the 36 wheat cultivars, while they exhibited smaller variations in whole grains. For example, the C.V. of the total ARs on seed surface was 43.63%, which was much higher than that of whole grains, and the C.V. of the AR homologues on the seed surface also tended to be higher (Table A2 and Table A3).

On the other hand, the relative abundances of AR homologues in whole grains and seed surfaces were different. AR 23:0 was the predominant homologue on seed surfaces of all wheat samples (Appendix A), with large variations in the relative composition of other ARs, such as AR 19:0 (C.V. was 78.20%) and AR 21:0 (C.V. was 23.07%) (Table 1). In contrast, in whole grains, the relative content of ARs (%) remained within a small range for all wheat samples (Appendix A), such as AR 19:0 (C.V. was 6.01%) and AR 21:0 (C.V. was 3.88%) (Table 1). AR 17:0/AR 21:0 ratios in the 36 wheat samples ranged from 0.08 to 0.13, and the average AR 17:0/AR 21:0 ratio was 0.1 (Table A3), which was in accordance with previous reports [38]. Another notable result was that the average ratio of AR 19:0/AR 21:0 for the 36 wheat samples was 0.7 (C.V. was only 8.57%) (Table A3). Notably, the ratio between homologues was most stable in whole grains. There were small variations in the relative compositions of ARs in whole grains among different cultivars, but there were large variations in the total contents of ARs and their homologues. 

### 2.4. Correlation Analysis 

#### 2.4.1. Correlation Analysis between Homologues of ARs

Correlation analyses of ARs extracted from seed surfaces and whole grains have not been previously performed, and in many cases, marked differences in the compositions of samples from different environments and varieties have been reported [43,44,45,46]. To further our understanding of AR homologues, we calculated Spearman’s correlation coefficient and performed a significance test for the regression analysis. Based on mean values, Spearman’s correlation coefficients (r) of individual AR homologues and total ARs of all wheat samples were calculated separately (Table 2). As expected, a strong correlation (r = 0.974, *p* < 0.001) between the major homologue (AR 21:0) and total ARs was observed in whole grains. Furthermore, contents of all the homologues were well correlated, except for AR 15:0 (Table 2). On seed surfaces, a strong correlation (r = 0.932, *p* < 0.001) between the major homologue (AR 23:0) and total ARs was also observed. As expected, all the homologue contents correlated well with each other, except for AR 19:0 and AR 25:0; however, the reason for this correlation is unknown (Table 2).

#### 2.4.2. Correlation Analysis between AR Homologues in Whole Grains and that on Seed Surfaces

To determine the relationship between the AR content in whole grains and that on seed surfaces, we also performed correlations analysis between the two data sets. As shown in Table 3, there were significant correlations (*p* < 0.001) between these data, and a good correlation (r = 0.863, *p* < 0.001) between total AR content in whole grains and seed surface was confirmed (Table 3). At the same time, good correlations were also evident between AR homologues on seed surfaces and in whole grains, especially between AR 23:0 on seed surfaces and AR 21:0 in whole grains (r = 0.760, *p* < 0.001). 

#### 2.4.3. Predictive Model (Linear Regression Equations) Construction

Based on the above results, linear regression equations were calculated (Figure 6). Figure 6b shows a good linear (R^2^ = 0.683, *p* < 0.001) relationship between total ARs on seed surfaces and that in whole grains. We could predict total ARs concentrations in whole grains based on linear regression equation. There were very good linear relationships between AR 23:0 on seed surfaces and AR 21:0 in whole grains (R^2^ = 0.524, *p* < 0.001) (Figure 6a). The above homologues were the major components of ARs on seed surfaces and in whole grains. Therefore, analyzing AR 23:0 content on seed surface could predict AR 21:0 content in whole grains. These results showed that the linear regression equation y = 50.114x+200.708 and y = 45.548x+392.343 could be used as models to predict ARs or AR 21:0 content in whole grains.

### 2.5. Methods Comparison

To validate which linear regression equation (predictive model) was feasible for predicting AR content in whole grain flour, agreements between ARs (AR 21:0) content of whole grain samples determined by GC and predictive models were found (Figure 7). Two methods are said to agree if they have the same bias and variance is homoscedasticity [47].

As shown in Figure 7a, total ARs content in whole grains determined by GC was linearly regressed as independent variable against total ARs content in whole grains determined by the predictive model. As expected, they displayed a good liner relationship (R^2^ = 0.826, *p* < 0.001). At the same time, the Bland–Altman analysis showed that more than 92% of variance points fell in the 95% confidence intervals; thus, the two methods were said to agree. Although there was a weak trend (y = 0.23x − 142.1, *p* < 0.001) showing increased difference between the two methods (Figure 7c), this increase in difference is of small practical importance and in most cases the error will be smaller than the analytical precision. Similarly, Figure 7b,d also revealed that there was no difference in AR 21:0 content determined by GC and the predictive model. The above results prove that the predictive models can be applied for predicting ARs content in whole grain flour.

### 2.6. Wheat Germination Results

In this study, we extracted lipids from seed surface using chloroform in our study, and the structure of wheat seed was not destroyed. Seed germination tests were performed on three wheat varieties (YM1, HM608 and LY3 were chosen randomly from the 36 wheat cultivars). The results showed that there were statistically nonsignificant differences in the germination rates of the normal seeds compared to the chloroform-soaked seeds (Figure 8). Therefore, this method was also economical. During the early stages of the breeding process, only small quantities of grain are available. In our study, the chloroform extraction method could meet this need, which requires only a small number of seeds to determine ARs contents, and after extraction, wheat seeds can still germinate normally. It was also clear that chloroform could be used to complement the non-destructive chemical analysis, providing information on ARs in cereal grains without the need for sophisticated chemical methods.

## 3. Discussion

This is the first time quantitative analysis of the lipids in whole grains and seed surfaces is reported. Furthermore, correlation analyses were conducted to determine the relationship between ARs on seed surfaces and in whole grains.

Both content of total ARs and AR homologues on seed surfaces and that in whole grains varied dramatically among the 36 wheat cultivars, while the relative composition of AR homologues in whole grains showed smaller variations. In whole grains, the average AR 17:0/AR 21:0 ratio was 0.1. Ratios of 0.1, 0.01 and 1.0 were reported for common wheat, durum wheat, and rye, respectively. These ratios have been suggested as an index for determining the source of cereal products [46,48]. Our study demonstrates that all tested samples were common wheat rather than rye or durum wheat. In fact, variations in ARs, AR 17:0/AR 21:0 ratio, and AR 19:0/AR 21:0 ratio were very low compared to the general variability of plant secondary metabolites [45], which supports their use as biomarkers. This study also indicates that minor amounts of ARs were present on seed surfaces (in the outer cuticle of pericarp), as suggested by Landberg et al. [21], meaning that the majority of ARs were present in the intermediate layer of seeds. Based on the above discussion, ARs can be used as a selective marker of the intermediate layer or testa in cereal fractions.

A low AR content was present on seed surfaces, and it showed a different homologue composition, indicating that AR compositions were different in different parts of wheat seed. Dayan et al. suggested that the formation of ARs with different chain lengths is highly dependent on the specificity of the appropriate enzyme for the substrates [49,50]. We speculated that the reason for these differences was that the enzyme activities were not equal in and outside the pericarp of wheat seeds. Although ARs have been found in an increasing number of organisms [28,37,38,51], a broader understanding of their bioactivities and underlying mechanisms is lacking. Further studies should be conducted to elucidate the related enzymes and substrates. More physiological experiments are needed in future studies.

On the other hand, the chloroform extraction procedure is a time-saving and simple extraction method, which can meet the need to efficiently screening large numbers of lines. Without affecting the seed germination rate and damaging seed structure, this method is suitable for breeders and nutritionists.

In summary, in this work, we analyzed the composition of SSL and WGL. Although ARs were the major compounds in both SSL and WGL, there were many differences between their compositions. Moreover, there was a highly significant correlation (*p* < 0.001) between total AR concentrations in whole grains and that on seed surfaces. Along with a substantially shorter sample extraction time, the major advantage of this method was that using chloroform allowed easy and non-destructive extraction of the ARs from seed surfaces. Therefore, results from this study could provide a non-destructive extraction method for determining ARs contents on seed surfaces, so as to predict the contents in whole grain flour. Using the methods developed in this paper can therefore help breeders select appropriate recombination events to accelerate the selection process without destroying seed structure or affecting seed germination rates.

## 4. Materials and Methods 

### 4.1. Plant Materials and Reagents 

Thirty-six wheat samples collected from different parts of China were kindly provided and identified by Crop Molecular Biology and Breeding laboratory, College of Agronomy, Northwest A&F University, China. The corresponding information of 36 wheat samples are listed in Appendix B
Table A1. All wheat samples were grown in the field during the 2017–2018 wheat-growing seasons in China. A total of 90 seeds per variety were individually hand-planted in a 1.5-m row at 10-cm spacing. Standard cultural practices for wheat were followed during the cultivation. All wheat materials were harvested in June 2018. 

Chloroform and ethyl acetate were purchased from Xilong Scientific (Shantou, China) and used for lipid extraction from wheat samples. Pyridine, n-tetracosane (C24) and *N*,*O*-bis (trimethylsilyl)- trifluoroacetamide (BSTFA) were purchased from Sigma-Aldrich (St. Louis, MO, USA). GC-MS (GCMS-QP2010, SHIMADZU, Tokyo, Japan), and GC-FID (GC-2010 PLUS, SHIMADZU, Tokyo, Japan) were applied for identifying wheat lipid components.

### 4.2. Developing a Suitable Method for Extracting Lipids from Seed Surfaces

Surface lipids were operationally defined as the lipid material extracted by quickly immersing these organs in chloroform [27,37,39]. In this paper, different extraction times were tested (10 s, 30 s, 60 s, 90 s, 120 s,180 s, 300 s, 600 s, and 1800 s), and it was found that SSL were fully extracted after 4 min (240 s) (Figure 1). The final protocol used was as follows: Wheat seeds (3 g) were immersed in a glass beaker containing 20 mL chloroform and 10 µg C24 as an internal standard, then shaken four times for 4 min at room temperature. Each lipid sample was filtered through a paper filter, then transferred to a GC autosampler vial and dried under a stream of nitrogen gas.

### 4.3. Isolation of Lipids from Whole Grain Flour

The sample extraction procedure followed a previously established method for grains [21,46,47]. Ross et al. confirmed that lipids could be extracted completely from whole grain flour in 24 h [27]. The extraction programs were as follows: Wheat seeds were air-dried and knife-milled, then stored in polyethylene pouches at −20 °C before extraction. Grain samples (3 g) were immersed in a glass beaker containing 40 mL ethyl acetate and 40 µg C24 (internal standard); lipid was extracted by continuous stirring for 24 h at room temperature. Each lipid sample was filtered through a paper filter and dried under a stream of nitrogen gas.

### 4.4. Derivatization Reactions 

For GC analysis, each lipid sample was derivatized with 40 μL pyridine and 40 μL BSTFA for 1 h at 70 °C, and each lipid sample was vortexed every 20 min [33,37,39]. The purpose of this step was to transform hydroxyl (OH-) containing compounds into their corresponding trimethylsilyl derivatives [33]. Then, the sample was dried under nitrogen gas. After that, 700 μL of chloroform was added for the next analysis. 

### 4.5. Chemical Analysis of Lipids 

After derivatization, lipid composition was analyzed on a capillary GC column (30 m long, i.d. = 0.32 mm, df = 0.25 μm; Restek, Bellefonte, PA, USA) and attached to an MS; helium (He) was used as carrier gas at a rate of 1.5 mL/min, programmed to temperature as follows: Set at 50 °C for 2 min, ramp 20 °C/min to 220 °C, hold for 2 min, ramp 1.6 °C/min to 310 °C, and hold for 18 min at 310 °C. The MS operated in full-scan mode applying a mass in the range of 35 to 700 *m*/*z*. Compounds were identified by comparing their retention times and mass spectra with authentic standards and literature data [36,38,39,40]. 

The GC equipped with a flame ionization detector (FID, SHIMADZU, Tokyo, Japan) was used for composition quantitative analysis. GC-FID was carried out on the same GC conditions as above, but with Nitrogen (N_2_) carrier gas inlet pressure regulated for a constant flow of 2.0 mL/min. Individual lipid compounds were quantified against the C24 (internal standard) by automatically integrating peak areas. All values were reported on a dry matter (DM) basis. DM was determined by drying samples in an oven at 105 °C overnight, cooling in a desiccator, and then weighing. All wheat samples were analyzed in triplicate; the coefficient of variance (C.V.) was less than 10% for the three samples.

### 4.6. Wheat Germination Tests

YM1, HM608, and LY3 were chosen randomly from the 36 wheat cultivars to perform the germination tests. Both the normal seeds and the chloroform-soaked (5 min) seeds were in the same germination conditions. The germination test was performed in a petri dish with two layers of filter papers moistened with distilled water, at 25 °C in the dark. Each petri dish contained 100 seeds, 5 mL of sterilized water was added to meet the conditions of germination humidity, and the number of statistical seed germinations was observed every day. A seed was regarded as germinated when the radicle had pierced the seed coat. The germination rate was counted on the 7th day. The results presented here are the means of the germination rates obtained from four replicates; the coefficient of variance (C.V.) was less than 10% for the four samples.

### 4.7. Statistical Analysis 

All results in our study denote the means ± SD. Pictures were drawn using Sigma plot 14.0 software, except for chemical structures, which were drawn using ChemDraw Professional 15.1 software. Correlation and linear analyses were performed using SPSS 19.0 software. Method agreement was assessed by a Bland–Altman plot [52]. A *p*-value less than 0.05 was considered to be statistically significant. 

## Figures and Tables

**Figure 1 molecules-24-01329-f001:**
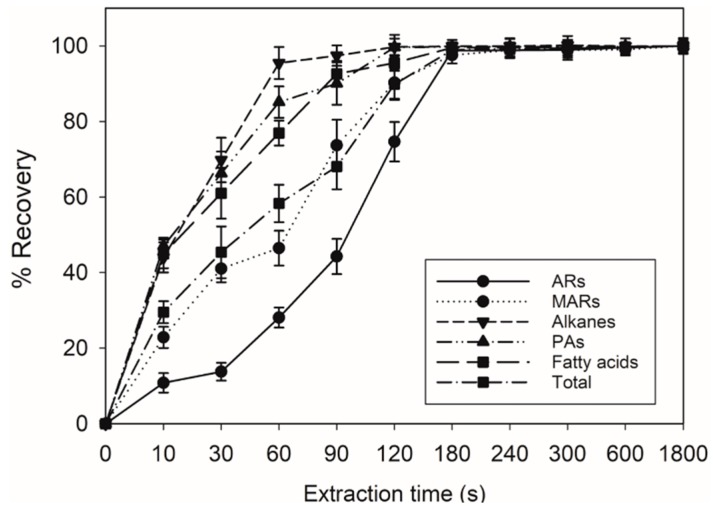
Recovery of total lipids and individual lipid components by chloroform extracting of the wheat seed surface. ARs, *n*-alkylresorcinols; MARs, *n*-methylalkylresorcinols; PAs, primary alcohols. Data points are expressed as the mean ± SD of four biological replicates.

**Figure 2 molecules-24-01329-f002:**
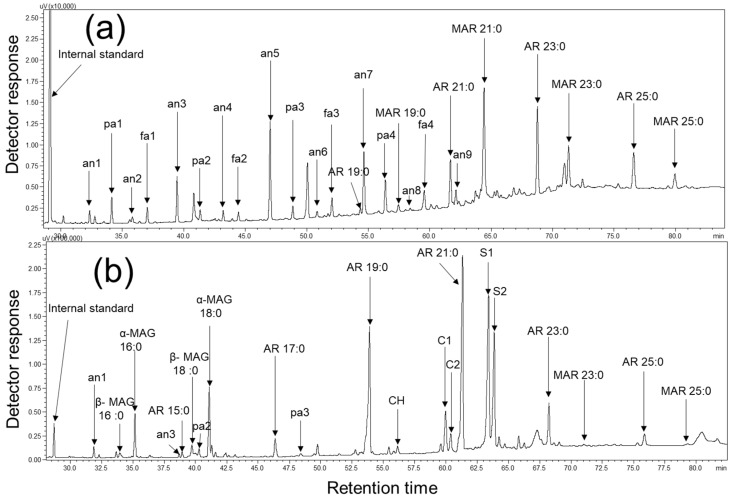
GC chromatogram of SSL (**a**) and WGL (**b**) of Houdemai 981 (HDM981). Internal standard, tetracosane (C24); AR(n): n-alkylresorcinols; MAG(n): n-monoacylglycerols; MAR(n): n-methylalkylresorcinols; n denotes the carbon atom number. Other compounds reflected are as follows: an1, pentacosane; an2, hexacosane; an3, heptacosane; an4, octacosane; an5, nonacosane; an6, triacontane; an7, hentriacontane; an8, dotriacontane; an9, tritriacontane; pa1, docosanol; pa2, tetracosanol; pa3, hexacosanol; pa4, octacosanol; fa1, docosanoic acid; fa2, tetracosanoic acid; fa3, hexacosanoic acid; fa4, octacosanoic acid; CH, cholesterol; C1, campesterol; C2, campestanol; S1, sitosterol; S2, sitostanol.

**Figure 3 molecules-24-01329-f003:**
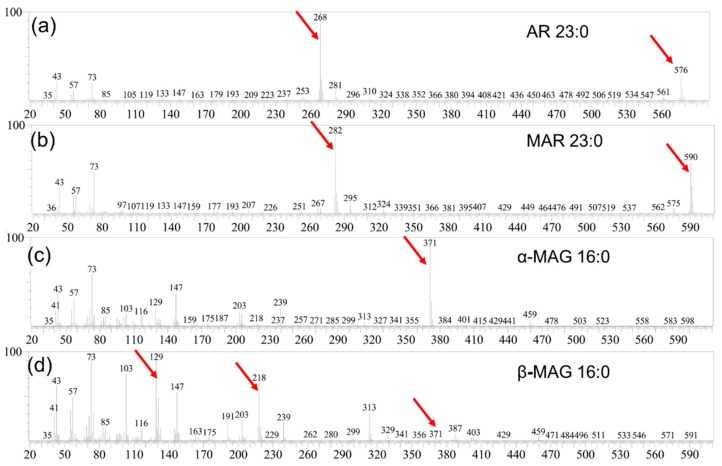
Identification of saturated ARs, MARs and MAGs present in the lipids by GC-MS of their bis-TMSi derivatives: mass spectra of AR 23:0 (**a**), MAR 23:0 (**b**), α-MAG 16:0 (**c**) and β-MAG 16:0 (**d**).

**Figure 4 molecules-24-01329-f004:**
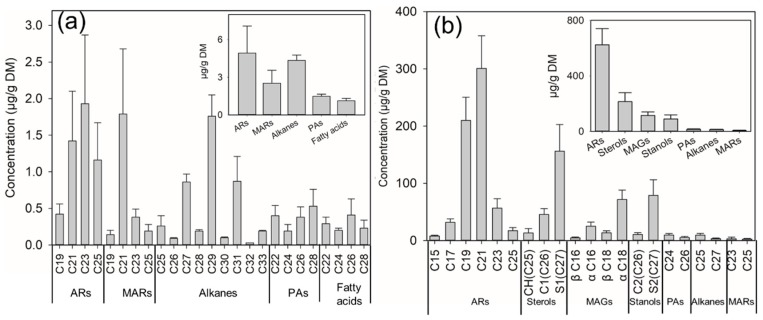
Composition and average content of SSL (**a**) and WGL (**b**) from 36 wheat samples. The ordinate shows carbon atom number. SD, standard deviation. The data are expressed as the mean ± SD (n = 36).

**Figure 5 molecules-24-01329-f005:**
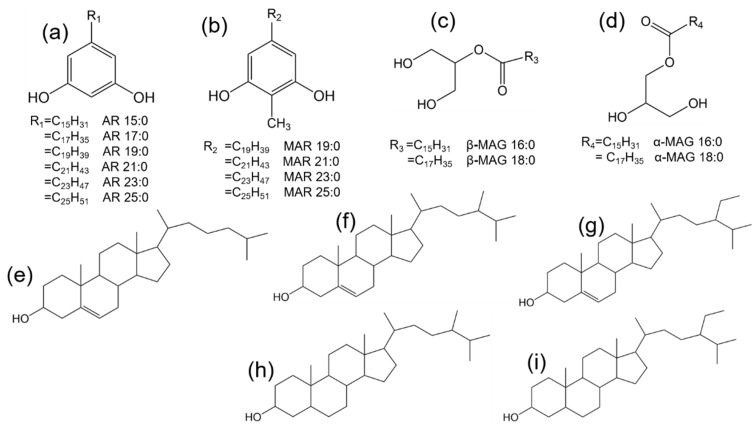
Structures and abbreviations representative of the main compounds identified in the extracts of wheat and referred in the text. (**a**) AR(n); (**b**) MAR(n); (**c**) β-MAG(n); (**d**) α- MAG(n); (**e**), cholesterol; (**f**) campesterol; (**g**) sitosterol; (**h**), campestanol; (**i**) sitostanol. R_n_ are n-alkyl chains.

**Figure 6 molecules-24-01329-f006:**
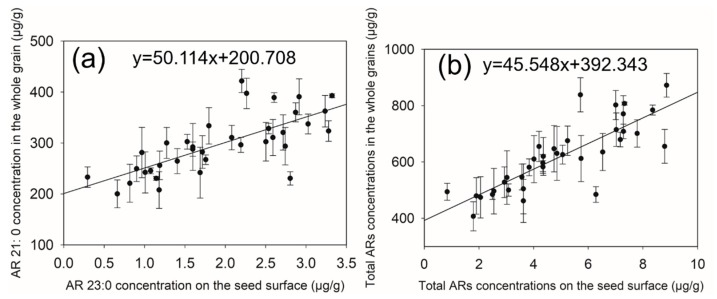
(**a**). Correlation between AR 23:0 concentration on seed surfaces and AR 21:0 concentration in whole grains. (R^2^ = 0.524) (**b**). Correlation between total ARs concentration on seed surfaces and that in whole grains (R^2^ = 0.683). R^2^, the coefficient of determination of the standard curve; in the graph, the linear equation of the standard curve is reported. Data points are expressed as the mean± SD of three biological replicates (n = 36, *p* < 0.001).

**Figure 7 molecules-24-01329-f007:**
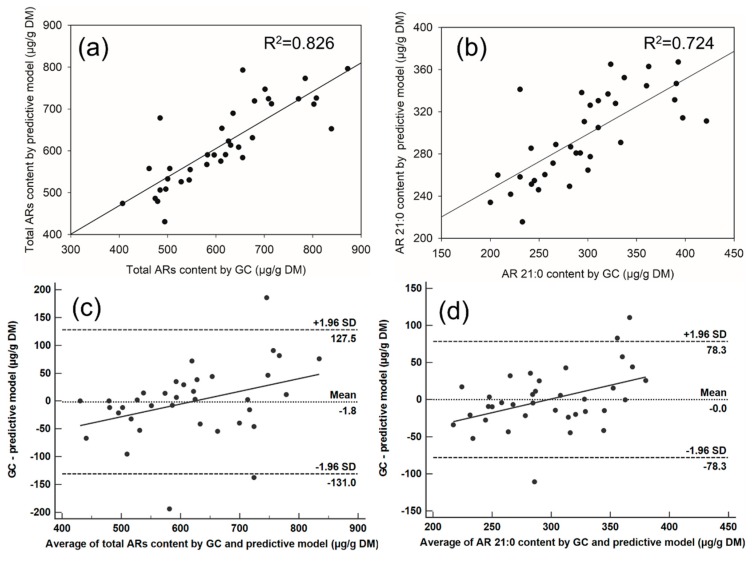
(**a**) AR 21:0 content in whole grain samples determined by GC and predictive model. y = 0.524x + 141.676 (n = 36, *p* < 0.001); (**b**) Total ARs content in whole grain samples determined by GC and predictive model. y = 0.683x + 195 (n = 36, *p* < 0.001). (**c**) and (**d**) The Bland–Altman plot illustrating the agreement between the predictive model-method and the GC-method. Average of the two methods is plotted on the x-axis and the observed difference between the methods on the y-axis (GC-predictive model), n = 36.

**Figure 8 molecules-24-01329-f008:**
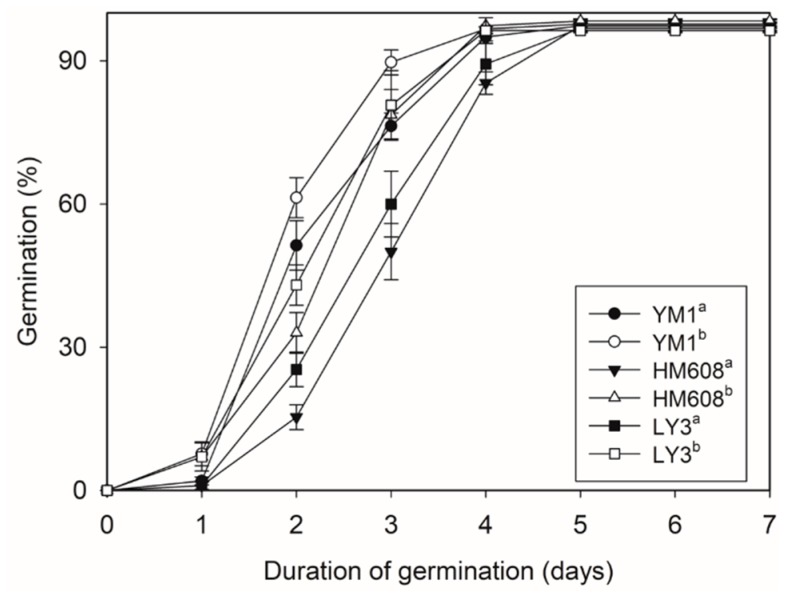
Effect of chloroform on wheat seed germination rates from three wheat samples (YM1, HM608 and LY3). ^a^ normal seeds (CK); ^b^ chloroform- soaked seeds (5 min). Data points are expressed as the mean ± SD of four biological replicates.

**Table 1 molecules-24-01329-t001:** Average relative proportions of AR homologue, 17:0/21:0 and 19:0/21:0 of 36 wheat samples cultivated in 2017 in China.

Samples	AR Homologue (%)							
	AR15:0	AR 17:0	AR 19:0	AR 21:0	AR 23:0	AR 25:0	17:0/21:0	19:0/21:0
Seed surface	nd	nd	7.35	27.47	39.83	25.35	nd	0.25
C.V. ^a^ (%)	nd	nd	78.20	23.07	17.50	31.20	nd	50.45
Whole grain	1.28	5.10	33.64	48.19	9.07	2.77	0.10	0.70
C.V. ^b^ (%)	24.19	**9.09**	**6.01**	**3.88**	17.11	20.29	11.44	**8.57**

nd, Not detected; C.V., Coefficient of Variance. ^a^, seed surface: ^b^, whole grain. Bold values indicate C.V. < 10%.

**Table 2 molecules-24-01329-t002:** Correlation coefficients between AR homologues content in whole grains and seed surfaces.

AR Homologue	AR Homologue					
	AR17:0	AR19:0	AR21:0	AR23:0	AR25:0	Total ARs
Whole grain						
AR 15:0	0.423 *	0.208	0.219	0.084	0.249	0.260
AR 17:0		**0.873** **	**0.830** **	0.571 **	0.610 **	**0.867** **
AR 19:0			**0.897** **	0.646 **	0.611 **	**0.945** **
AR 21:0				**0.711 ****	0.679 **	**0.974** **
AR 23:0					**0.815 ****	**0.765 ****
AR 25:0						**0.730 ****
Seed surface						
AR 19:0			**0.917** **	0.555 **	0.279	**0.799 ****
AR 21:0				**0.754** **	0.400 *	**0.932 ****
AR 23:0					**0.753** **	**0.855** **
AR 25:0						0.620 **

** means correlation significant at 99.99% confidence level, * means correlation significant at 99.95% confidence level. Bold values indicate correlation coefficients r > 0.7.

**Table 3 molecules-24-01329-t003:** Correlation coefficients between total ARs, AR homologue content on seed surfaces and that in whole grains.

AR Homologue	AR Homologue				
	AR19:0 ^a^	AR 21:0 ^a^	AR 23:0 ^a^	AR 25:0 ^a^	Total Ars ^a^
AR19:0 ^b^	0.591 **	**0.713 ****	0.698 **	0.642 **	**0.825** **
AR 21:0 ^b^	0.573 **	0.682 **	**0.760** **	**0.703** **	**0.813** **
AR 23:0 ^b^	0.390 *	0.551 **	0.629 **	0.440 **	0.622 **
AR 25:0 ^b^	0.534 **	0.612 **	0.658 **	0.498 **	0.644 **
Total Ars ^b^	0.627 **	**0.750** **	**0.775** **	0.676 **	**0.863** **

^a^ seed surface; ^b^ whole grain. ** means correlation significant at 99.99% confidence level, * means correlation significant at 99.95% confidence level. Bold values indicate correlation coefficients r > 0.7.

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
