# Peer review of "Non-Destructive Determination of Alkylresorcinol (ARs) Content on Wheat Seed Surfaces and Prediction of ARs Content in Whole-Grain Flour"

_molecules, 2019, doi:10.3390/molecules24071329_

Round 1

Reviewer 1 Report

The manuscript submitted by Wang et al. describes an interesting method for non-destructive determination of alkylresorcinols in wheat “Seed Surface Lipids”. Furthermore, they evidenced that this measurement is highly correlated with the AR content in grain, so that this approach could be of great interest to facilitate the rapid screening of these phytochemicals in wheat breeding programs.

The exprimental design, analysis and statistics seem well conducted. My concern is only related to the quality of the text (there are a lot of grammatical errors, non-informative and short sentences, lack of clarity in some parts, etc).

Before this manuscript can be considered for publication, a deep revision of the written English by a native speaker is mandatory.

Author Response

Dear Reviewer:

Thank you very much for your letter about our manuscript title of "A non-destructive and accurate approach to determine the alkylresorcinols (ARs) content on the wheat seed surface and predict ARs content in whole-grain flour (ID: molecules-474814)". We have carefully revised and supplemented the manuscript. The following is our response to each of your comments.

Point: The experimental design, analysis and statistics seem well conducted. My concern is only related to the quality of the text (there are a lot of grammatical errors, non-informative and short sentences, lack of clarity in some parts, etc.). Before this manuscript can be considered for publication, a deep revision of the written English by a native speaker is mandatory.

Response: We have tried our best to revise our manuscript according to your comments, and the relevant revisions were highlighted in red for clearness, which may provide a more readable description on the method and the main results of this study. The manuscript has been corrected by a native English speaker.

Attached please find the revised version, which we would like to submit for your kind consideration. If you have any questions about this manuscript, please don’t hesitate to let us know.

Best regards! 

Yours sincerely,

Jiahuan Wang

Reviewer 2 Report

Wang et al. have submitted an interesting study on a non-destructive approach for determionation of alkylresorcinols (ARs) content on the wheat seed surface and construction of predictive models to predict ARs content in whole-grain flour. Prior to acceptance of this manuscript, minor revisions are required as listed below.

The title is a tad overoptimistic. Words such as "accurate" ahoush generally be avoided in research articles. I propose to revise the title to:

Non-destructive determination of alkylresorcinols (ARs) content on the wheat surface and prediction of ARs conent in whole-grain flour

English proofing is needed. Several sentences have weird constructs; e.g., "In order to extract ARs from whole grains, the seeds always are ground, and their structures are damaged, few studies had reported on the seed surface lipids." There are many other examples in the text, professional proofing is advised.

In the abstract on lines 17-19, the authors state there is a positive correlation between average total contents of ARs. This is nonsense. Break the sentence into two and change "values" to "parameters": "There was a positive correlation (r = 0.863) between these two parameters". Also, statistical significance testing is a MUST in this case. Please do so, and report p values beside r.

Abstract; L19-21: change to: "Results suggest that..."

Abstract; L22: "up to date" ??

P2L53; P2L69: "no damaging" ??

After reading the Introduction section, it is still unclear to me what exactly is the importance of quantifying / predicting the content of ARs in whole grain flour from ARs on the seed surface, except it being a non-destructive approach. Please clarify this and justify your work. An overview of the state-of-the-art in this section would help this greatly.

Figure 1: there are no repeats? If yes, report error bars too. If there are no repeats, the experiment in this study is quite flawed.

Table 2: You should rename α to 1-α instead; i.e., 99.99 and 99.95 % confidence levels.

How were the compounds based on GC-MS spectra identified? Library search? If so, what was the threshold for identification?

P11L319: Report p values in parentheticals when referring to significant correlation.

What was the rationale of choosing GC-MS as the method of choice for analysis of lipids? HPLC-ESI-MS/MS was found to be an excellent choice for analysis, quantification and identification of (phospho)lipids from biological (and food) samples (Buszewski, B.; Walczak, J.; Žuvela, P.; Liu, J. J. J. Chromatogr. A 2017, 1487, 179-186.).

Author Response

Dear Reviewer:

Thank you very much for your letter about our manuscript title of "A non-destructive and accurate approach to determine the alkylresorcinols (ARs) content on the wheat seed surface and predict ARs content in whole-grain flour (ID: molecules-474814)". We have carefully revised and supplemented the manuscript. The following is our response to each of your comments.

Point 1: The title is a tad overoptimistic. Words such as "accurate" ahoush generally be avoided in research articles. I propose to revise the title to: Non-destructive determination of alkylresorcinols (ARs) content on the wheat surface and prediction of ARs content in whole-grain flour.

Response 1: According to your recommendation, we have revised the title to: Non-destructive determination of alkylresorcinols (ARs) content on wheat seed surfaces and prediction of ARs content in whole-grain flour

Point 2: English proofing is needed. Several sentences have weird constructs; e.g., "In order to extract ARs from whole grains, the seeds always are ground, and their structures are damaged, few studies had reported on the seed surface lipids." There are many other examples in the text, professional proofing is advised.

Response 2: The manuscript has been corrected by a native English speaker, and the relevant revisions were highlighted in red for clearness.

Point 3: In the abstract on lines 17-19, the authors state there is a positive correlation between average total contents of ARs. This is nonsense. Break the sentence into two and change "values" to "parameters": "There was a positive correlation (r = 0.863) between these two parameters". Also, statistical significance testing is a MUST in this case. Please do so, and report p values beside r.

Response 3: According to your recommendation, we have revised this sentence to: The average total content of ARs (624.223 µg/g) in whole grains was much higher than that on seed surfaces (4.934 µg/g), with a positive correlation (r = 0.863, p<0.001) between these two parameters.

Point 4: Abstract; L19-21: change to: "Results suggest that..."

Response 4: We have revised this sentence.

Point 5: Abstract; L22: "up to date" ??

Response 5: We have deleted "up to date", and revised this sentence to: we demonstrate that a non-destructive extraction approach is an appropriate and useful method

Point 6: P2L53; P2L69: "no damaging" ??

Response 6: "no damaging" was changed to " non-destructive"

Point 7: After reading the Introduction section, it is still unclear to me what exactly is the importance of quantifying / predicting the content of ARs in whole grain flour from ARs on the seed surface, except it being a non-destructive approach. Please clarify this and justify your work. An overview of the state-of-the-art in this section would help this greatly.

Response 7: Alkylresorcinols (ARs) are biomarkers for whole-grain cereals intake, there were many kinds of methods for extracting ARs from whole-grain cereals, but few of them are rapid, simple, and non-destructive. For example, ethyl acetate and acetone are the most commonly extraction solvents, but they always require long extraction times (6 to 24 h)[1-3]; Geerkens et al. (2015) had developed an ultrasound-assisted extraction method, this method was fast, but it required complicated operations, and high chemical-analytics costs [4]. After extraction, cereal samples were not recycled [1-4]. Overall, all the present AR extraction methods are not simple, cost-effective and non-destructive [1-4]. At the same time, the present methods are not helpful and convenient for breeders.

Therefore, developing a non-destructive method that can help breeders efficiently and quickly analyze ARs in large populations of wheat materials is necessary. Chloroform extraction method is fast, simple-operation, and highly-efficient [5-8]. Thus, this approach (predicting the content of ARs in whole grain flour from ARs on seed surface) was non-destructive, simple, rapid, and cost-effective.

1.     A. B. Ross et al. Food Chemistry 220, 344-351 (2017).

2.     J. Liu et al., Journal of Agricultural and Food Chemistry 66, 9241-9247 (2018).

3.     P. Prinsen et al. Journal of Agricultural and Food Chemistry 62, 1664-1673 (2014).

4.     C. H. Geerkens et al. Food Chemistry 169, 261-269 (2015).

5.     R. C. Racovita et al. Phytochemistry 130, 182-192 (2016).

6.     N. M. Adamski et al. Plant Journal 74, 989-1002 (2013).

7.     Y. Wang et al. Plos One 10 (2015).

8.     M. Wang et al. Frontiers in Plant Science 8 (2017).

We have clarified and justified its importance in the introduction section.

Point 8: Figure 1: there are no repeats? If yes, report error bars too. If there are no repeats, the experiment in this study is quite flawed.

Response 8: Yes, there are four biological replicates. We have drawn error bars in the Figure 1.

Point 9: Table 2: You should rename α to 1-α instead; i.e., 99.99 and 99.95 % confidence levels.

Response 9: We have revised this sentence to: ** means correlation significant at 99.99% confidence level, * means correlation significant at 99.95 % confidence level.

Point 10: How were the compounds based on GC-MS spectra identified? Library search? If so, what was the threshold for identification?

Response 10: Yes, the compounds were identified by library search, the threshold for identification was 95%.

Point 11: P11L319: Report p values in parentheticals when referring to significant correlation.

Response 11: We have reported the p values in parentheticals.

Point 12: What was the rationale of choosing GC-MS as the method of choice for analysis of lipids? HPLC-ESI-MS/MS was found to be an excellent choice for analysis, quantification and identification of (phospho)lipids from biological (and food) samples (Buszewski, B.; Walczak, J.; Žuvela, P.; Liu, J. J. J. Chromatogr. A 2017, 1487, 179-186.).

Response 12: A challenge in my work was to get adequate chromatographic separation of the different alkylresorcinols (ARs), methylalkylresorcinols (MARs), alkanes, primary alcohols, and fatty acids. Although HPLC- MS had been used to identification of ARs from cereal grains in some studies [1-3], Ross et al. (2017) found that HPLC did not give adequate separation of the different homologues of ARs and MARs in Quinoa (Chenopodium quinoa) [4], and that it was necessary to use GC to get adequate separation of the different homologues, and a MS to differentiate between the ARs and MARs. HPLC could be used to identify compounds containing special groups (C=C, C=O, C=N, etc.), however, wheat seed surface lipid contains some amounts of alkanes (C25-C33), HPLC did not give adequate separation of the different homologues, too.

Prinsen et al. (2014) identified more than 120 chemical compounds (including ARs, alkanes, free fatty acids, steroids, etc.) from wheat bran lipids by GC-MS [5]. GC-MS also have been widely applied in quantification and identification of surface lipids or waxes [6-11].

1.     J. Liu et al., Journal of Agricultural and Food Chemistry 66, 9241-9247 (2018).

2.     A. B. Ross et al. Journal of Agricultural and Food Chemistry 57, 5187-5193 (2009)

3.     C. H. Geerkens et al. Food Chemistry 169, 261-269 (2015).

4.     A. B. Ross et al. Food Chemistry 220, 344-351 (2017).

5.     P. Prinsen et al. Journal of Agricultural and Food Chemistry 62, 1664-1673 (2014).

6.     R. C. Racovita et al. Phytochemistry 130, 182-192 (2016).

7.     N. M. Adamski et al. Plant Journal 74, 989-1002 (2013).

8.     Y. Wang et al. Plos One 10 (2015).

9.     M. Wang et al. Frontiers in Plant Science 8 (2017).

10.   X. Ji et al. Phytochemistry 2008, 69, 1197-1207 (2008)

11.  M. Wen et al. Journal of Experimental Botany 2009, 60, 1811-1821 (2009)

Therefore, we choose GC-MS to analyze seed surface and whole grain lipids.

If you have any questions about this manuscript, please don’t hesitate to let us know.

Best regards! 

Yours sincerely,

Jiahuan Wang

Round 2

Reviewer 1 Report

The manuscript is currently acceptable for publication